organic chemistry/medicinal chemistry

c-PLAI, cyclotetrapeptide, antimicrobial peptide, solid-phase peptide synthesis

**Authors for correspondence:**
Rani Maharani
e-mail: r.maharani@unpad.ac.id
Koichi Fukase
e-mail: koichi@chem.sci.osaka-u.ac.jp

This article has been edited by the Royal Society of Chemistry, including the commissioning, peer review process and editorial aspects up to the point of acceptance.

# Synthesis of cyclotetrapeptide analogues of c-PLAI and evaluation of their antimicrobial properties

Rani Maharani[1,2], Orin Inggriani Napitupulu[1],
Jelang M. Dirgantara[1], Ace Tatang Hidayat[1,2],
Dadan Sumiarsa[1], Desi Harneti[1], Nurlelasari[1],
Unang Supratman[1,2] and Koichi Fukase[3]

[1]Laboratorium Sentral, and [2]Department of Chemistry, Faculty of Mathematics and Natural Sciences, Universitas Padjadjaran, Jalan Raya Bandung Sumedang Km 21, Jatinangor, Kabupaten Sumedang, 45363 West Java, Indonesia
[3]Graduate School of Science, Osaka University, Toyonaka, Osaka 560-0043, Japan

RM, 0000-0001-8156-9773; KF, 0000-0001-8844-0710

Antimicrobial peptides (AMPs) are interesting compounds owing to their ability to kill several pathogens. In order to identify new AMPs, c-PLAI analogues were synthesized and evaluated together with their linear precursors for their antimicrobial properties against two Gram-positive bacteria (*Staphylococcus aureus* and *Bacillus cereus*), two Gram-negative bacteria (*Escherichia coli* and *Klebsiella pneumoniae*), and two fungal strains (*Candida albicans* and *Trichophyton mentagrophytes*). The new c-PLAI analogues were prepared through a combination of solid- and solution-phase syntheses, as previously employed for the synthesis of c-PLAI. The antimicrobial activity tests showed that the synthetic parent peptide c-PLAI was inactive or weakly active towards the bioindicators employed in the assay. The tests also indicated that cyclic c-PLAI analogues possessed enhanced antimicrobial properties against most of the bacteria and fungi tested. Furthermore, this study revealed that analogues containing cationic lysine residues displayed the highest activity towards most bioindicators. A combination of lysine and aromatic residues yielded analogues with broad-spectrum antimicrobial properties.

## 1. Introduction

c-PLAI [cyclo(pro-leu-ala-ile)] is a cyclotetrapeptide that was isolated by Rungprom and co-workers in 2008 from marine

**Figure 1.** Structures of c-PLAI and its analogues.

*Pseudomonas* sp., which was associated with the Japanese seaweed *Diginea* sp. [1]. All the amino acid residues of c-PLAI were reported to be L-configured. The authors stated that c-PLAI was inactive as an antibacterial agent against both Gram-negative and Gram-positive bacteria, namely *Staphylococcus aureus*, *Micrococcus luteus*, *Bacillus subtilis*, *Escherichia coli* and *Vibrio anguillarum*. Later, c-PLAI was successfully synthesized using a convergent solution-phase method [2]. The synthetic product was reported to possess antimicrobial properties with a minimum inhibitory concentration (MIC) value of 6 µg ml$^{-1}$ against two Gram-negative bacteria, i.e. *Pseudomonas aeruginosa* and *Klebsiella pneumoniae*, as well as three fungal strains, i.e. *Microsporum andouinii*, *Candida albicans* and *Trichophyton mentagrophytes*. In view of the inconsistencies of these reports, we planned to explore the antimicrobial properties of synthetically produced c-PLAI and its analogues.

Recently, our group described total synthesis of c-PLAI through a combination of solid- and solution-phase methods [3]. The proposed synthetic route was found to be more efficient than that previously reported by Dahiya & Gautam [2] owing to the use of a faster solid-phase peptide synthesis for the generation of the linear precursor and a more effective cyclization step. This synthetic protocol was used to synthesize several analogues of c-PLAI in order to discover new antimicrobial peptides (AMPs) with enhanced antimicrobial properties.

All analogues (figure 1) were designed to retain proline in their structure, as several groups reported that proline is a key amino acid residue in various AMPs [4,5]. Proline has also been known to promote β-turn structure that favours cyclization [6,7]. Further modifications were introduced by replacing the amino acid leucine or alanine of c-PLAI with hydrophobic (valine, phenylalanine) or cationic (lysine) amino acids, thus resulting in five analogues, i.e. c-PLPI [cyclo(pro-leu-pro-ile)], c-PLVI [cyclo-(pro-leu-val-ile)], c-PLFI [cyclo-(pro-leu-phe-ile)], c-PKAI [cyclo-(pro-lys-ala-ile)] and c-PKFI [cyclo-(pro-lys-phe-ile)]. Increased hydrophobicity and cationicity were reported to enhance the antimicrobial properties of peptides [8–12]. Isoleucine was maintained in the structure because isoleucine is known to be more hydrophobic than alanine and leucine. Furthermore, c-PLAI and its analogues including their linear precursors were evaluated against two Gram-positive bacteria, namely *Staphylococcus aureus* and *Bacillus cereus*, two Gram-negative bacteria, namely *Escherichia coli* and *Klebsiella pneumoniae*, and two fungal strains, namely *Candida albicans* and *Trichophyton mentagrophytes*.

In the present study, the antimicrobial properties of c-PLAI and its analogues were evaluated, along with those of their linear peptide precursors.

# 2. Experimental section

All the compounds were analysed using the analytical RP-HPLC Waters 2998 Photodiode Array Detector, LiChrospher 100 RP-18 5 mm column. The purification took advantage of Buchi C-620

Sepacore with C18 column (12 g). $^1$H- and $^{13}$C-NMR spectra were recorded on Agilent NMR 500 MHz ($^1$H) and 125 MHz ($^{13}$C) using deuterated solvent, JEOL NMR 600 MHz ($^1$H) and 150 MHz ($^{13}$C), and BRUKER NMR 600 MHz ($^1$H) and 150 MHz ($^{13}$C). Mass spectrometry spectra were recorded on Waters HR-TOF-MS Lockspray. Loading resin absorbance was measured using the TECAN Infinite pro 200 UV–Vis Spectrophotometer. All of the amino acid residues, Fmoc-L-proline, Fmoc-L-leucine, Fmoc-L-alanine, Fmoc-L-Isoleucine, Fmoc-L-phenylalanine, Fmoc-L-valine, Fmoc-L-Lysine (Boc) and 2-chlorotrityl chloride resin (0.972 mmol g$^{-1}$) were purchased from GL-Biochem Ltd., China. Dichloromethane, *N,N*-dimethylformamide (DMF), trifluoroethanol, acetic acid and trifluoroacetic acid used in the synthesis were of analytical grade.

## 2.1. Synthesis and purification of linear tetrapeptides

Linear tetrapeptides were synthesized using the manual solid-phase peptide synthesis procedure, with 2-chlorotrityl chloride resin employed as the solid support. 5 ml of dichloromethane was added into 2-chlorotrityl chloride resin (400 mg, 0.39 mmol) and the resin was shaken with a rotary suspension mixer for 5 min. Dichloromethane was removed from the reactor by filtration using an air pump. The first amino acid (AA$_1$) (0.9 mmol) in 5 ml of dichloromethane and DIPEA (387.2 µl; 2.22 mmol) was loaded into the resin for 24 h at room temperature. The resin was dried with an air pump after the resin loading. The loading resin value (amount of amino acid loaded in 1 g of resin) was determined by deprotecting the Fmoc group of the first amino acid attached to 0.5 mg resin by 20% piperidine in DMF for 1 h. The total amount of Fmoc deprotected from the amino acid was quantified using Spectrophotometer UV–Vis, with a wavelength of 290 nm. To calculate the resin loading value, the absorbance was inserted into a formula: Loading (mmol g$^{-1}$) = (Abs$_{resin}$)/(mg of resin × 1.75) [13]. The total amount of Fmoc equals the total number of amino acids loaded on the resin. Afterwards, the resin was capped with 10 ml solution of MeOH : DIPEA : DCM (1.5 : 0.5 : 8). After the resin had been capped, Fmoc group of the first amino acid was deprotected using 20% piperidine in DMF for 30 min. Following this, the resin was filtered and washed using 3 × 3 ml DMF and 3 × 3 ml dichloromethane. The success of the Fmoc deprotection was analysed via a chloranil test. If the test gave the result of green or red resin after 5 min of reaction, this indicated that the deprotection was successful, but if the test gave the result of yellow resin, this indicated that the deprotection was unsuccessful and the deprotection needed to be repeated. After the Fmoc deprotection step, the solution of the second amino acid (AA$_2$) (2 equiv.), HATU (2 equiv.), HOAt (2 equiv.) and DIPEA (8 equiv.) in 5 ml of dichloromethane : DMF (1 : 1) was coupled with the first amino acid on the resin using a rotary suspension mixer for 24 h. Subsequently, the resin was filtered and washed using 3 × 3 ml DMF and 3 × 3 ml dichloromethane. The success of the amino acid coupling was analysed via a chloranil test. If the test gave the result of yellow resin after 5 min of reaction, this indicated that the coupling was successful, but if the test gave the result of green or blue resin, this indicated that the coupling was unsuccessful and the coupling needed to be repeated. The repetitive deprotection and coupling process was carried out until the desired tetrapeptide attached to the resin. Tetrapeptide on the resin was cleaved using 5 ml of 5% of TFA in dichloromethane for 30 min for PLPI, PLVI and PLFI or 5 ml of AcOH/TFE/DCM (2 : 2 : 6) for PKAI and PKFI. The solution was collected and evaporated for further purification. Crudes of PLPI, PLVI, PLFI, PKAI and PKFI were purified using preparative RP-HPLC with Agilent Pursuit 5 C-18 (250 × 21.2 mm, 5 µl) with eluent of water/ acetonitrile (5–95%) for 60 min, and flow rate of 3.5–5.0 ml min$^{-1}$. Each fraction was collected and concentrated with a rotary evaporator to give purified peptides of PLPI, PLFI, PKAI and PKFI.

## 2.2. Cyclization of linear tetrapeptides and purification of cyclic tetrapeptides

HATU (2 equiv.) was dissolved in 300 µl of DMF, while linear tetrapeptide was dissolved in 10 ml of dichloromethane. Following this, the solution of HATU was added into the solution of tetrapeptide. DIPEA (3 equiv.) was added slowly into the mixture. The mixture of tetrapeptide, HATU and DIPEA was diluted with 50 ml of dichloromethane. The mixture was stirred at 0°C constantly for 1 h and then stirred at room temperature for 30 min. The reaction was monitored via thin layer chromatography using silica GF$_{254 \text{ nm}}$ (*n*-hexane:ethyl acetate 1 : 1). The reaction mixture was concentrated using a rotary evaporator after the reaction had finished. Crude cyclotetrapeptide was dissolved in 5 ml of chloroform and washed with 1 M HCl (3 × 25 ml), saturated NaHCO$_3$ (3 × 25 ml) and saturated NaCl (3 × 25 ml), respectively. The organic layer was then concentrated using a rotary evaporator. c-PLVI was obtained as a pure compound and was not further purified. Other crude peptides were purified using a column

chromatography with silica gel $G_{60}$ as the stationary phase and isocratic *n*-hexane/ethyl acetate (1:1) as the mobile phase to give purified peptide c-PLPI, c-PLFI, c-PKAI and c-PKFI. The purity of the synthetic peptides was evaluated by analytical RP-HPLC with C-18 column ($4 \times 125$ mm, 5 µl) using eluent of water/acetonitrile (5–95%) for 30 min, and flow rate of 1 ml min$^{-1}$.

## 2.3. Deprotection of side chain protecting group of linear and cyclic tetrapeptides

The side chain protecting group of linear and cyclic peptides was deprotected with 95% TFA in water (1 µl of TFA/1 mg of peptide). The reaction mixture was stirred at a slow rate for 30 min. The reaction mixture was concentrated using a rotary evaporator and dissolved in 25 ml of chloroform before being washed using 25 ml of saturated NaHCO$_3$ and 25 ml of brine solution. The organic layer was dried with anhydrous MgSO$_4$ and the filtrate was collected. The filtrate was concentrated using a rotary evaporator. The crude solution was purified using a column chromatography with silica gel G60 as the stationary phase, and a gradient of *n*-hexane : ethyl acetate (50–100%) as the mobile phase. Each fraction was collected and concentrated using a rotary evaporator to give a pure substance containing unprotected linear and cyclic peptides.

**PLPI**: $[\alpha]_D^{28}$ − 50.1 (*c* 0.10, MeOH); HR-TOF-ESI-MS *m/z* [M-H]$^+$ 437.2753 (calculated *m/z* (437.2764) for C$_{22}$H$_{37}$N$_4$O$_5$; $^1$H-NMR (600 MHz, CD$_3$OD): $\delta_H$ (ppm) 4.64 (1H, t, *J* = 6.6 Hz, H-α), 4.51 (1H, dd, *J* = 8.4, 4.2 Hz, H-α), 4.30 (1H, d, *J* = 5.4 Hz, H-α), 4.28–4.25 (1H, m, H-α), 3.81–3.77 (1H, m, H-δ Pro), 3.63–3.59 (1H, m, H-δ Pro), 3.38–3.34 (1H, m, H-δ Pro), 3.24–3.28 (1H, m, H-δ Pro), 2.43–2.37 (1H, m, H-β Ile), 2.18–2.12 (1H, m, H-β Pro), 2.07–2.03 (2H, m, H-β Pro), 2.02–1.96 (4H, m, H-β Pro; H-γ Pro), 1.90–1.84 (1H, m, H-γ Pro), 1.77–1.70 (1H, m, H-β Leu), 1.61–1.59 (1H, m, H-β Leu), 1.55–1.47 (1H, m, H-γ Ile), 1.28–1.21 (1H, m, H-γ Ile), 1.00–0.88 (12H, m, H-δ, δ′ Ile; H-δ, γ′ Leu); $^{13}$C-NMR (150 MHz, CD$_3$OD): $\delta_C$ (ppm) 174.6 (COOH Ile), 174.1 (C=O Leu), 172.9 (C=O Pro), 169.9 (C=O Pro), 61.2 (C-α Ile), 60.8 (C-α Leu), 58.3 (C-α Pro 2x), 51.6 (C-δ Pro), 47.5 (C-δ Pro), 38.4 (C-β Ile), 30.9 (C-β Pro), 30.2 (C-β Pro), 26.2 (C-γ Pro), 26.0 (C-γ Pro), 24.9 (C-γ Ile), 23.8 (C-γ Leu), 21.6 (C-δ Leu), 21.5 (C-δ Leu), 16.1 (C-γ′ Ile), 11.9 (C-δ Ile).

**PLFI**: $[\alpha]_D^{27}$ − 9.3 (*c* 0.10, MeOH); HR-TOF-ESI-MS *m/z* [M + H]$^+$ 489.3078 (calculated *m/z* 489.3077 for C$_{26}$H$_{41}$N$_4$O$_5$; $^1$H-NMR (600 MHz, CD$_3$OD): $\delta_H$ (ppm) 7.22 (4H, d, *J* = 5.0 Hz, aryl-H Phe), 7.18–7.14 (1H, m, aryl-H Phe), 4.39 (2H, quint, *J* = 7.5 Hz, H-α), 4.32 (1H, d, *J* = 7.5 Hz, H-α), 4.27 (1H, t, *J* = 7.5 Hz, H-α), 3.38–3.35 (1H, m, H-β Phe), 3.32–3.30 (1H, m, H-β Phe), 3.13 (1H, d, *J* = 6.0 Hz, H-δ Pro), 2.89 (1H, t, *J* = 6.0 Hz, H-δ Pro), 2.35–2.28 (1H, m, H-β Ile), 2.01–1.84 (4H, m, H-β Pro; H-β Leu), 1.62–1.55 (1H, m, H-γ Pro), 1.48 (3H, t, *J* = 6,6 Hz, H-γ Pro; H-γ Leu; H-γ Ile), 1.26–1.17 (1H, m, H-γ Ile), 0.95–0.86 (12H, m, H-δ, γ′ Ile; H-δ, δ′ Leu); $^{13}$C-NMR (150 MHz, CD$_3$OD): $\delta_C$ (ppm) 173.3 (COOH Ile), 173.2 (C=O Leu), 172.4 (C=O), 168.4 (C=O), 136.2 (aryl Cq), 128.5 (aryl, CH meta), 127.0 (aryl, CH ortho), 120.7 (aryl CH, para), 59.5 (C-α 2x), 56.6 (C-α), 52.1 (C-α), 48.7 (C-δ Pro), 46.0 (C-β Leu), 40.5 (C-β Ile), 37.1 (C-β Phe), 29.6 (C-β Pro), 27.1, 24.7 (C-γ Pro), 24.5 (C-γ Ile), 23.5 (C-γ Leu), 22.0 (C-δ Leu), 20.4 (C-δ Leu), 16.6, 14.6 (C-γ′ Ile), 10.5 (C-δ Ile).

**PLVI**: $[\alpha]_D^{28}$ − 51.7 (*c* 0.10, MeOH); HR-TOF-ESI-MS *m/z* [M + H]$^+$ 441.3073 (calculated *m/z* 441.3077 for C$_{22}$H$_{41}$N$_4$O$_5$; $^1$H-NMR (600 MHz, CD$_3$OD): $\delta_H$ (ppm) 4.40 (2H, quint, *J* = 7.2 Hz, H-α), 4.31 (1H, d, *J* = 5.4 Hz, H-α), 4.27 (1H, t, *J* = 8.4 Hz, H-α), 3.40–3.35 (1H, m, H-δ Pro), 3.32–3.26 (1H, m, H-δ Pro), 2.42 (1H, sext, *J* = 8.4 Hz, H-β Val), 2.10–2.07 (1H, sext, *J* = 6.6 Hz, H-β Ile), 2.02–1.99 (2H, m, H-β Pro), 1.88–1.86 (1H, m, H-β Leu), 1.68 (2H, sext, *J* = 6.6 Hz, H-γ Pro), 1.60 (2H, t, *J* = 7.8 Hz, H-γ Ile), 1.50–1.46 (1H, m, H-γ Leu), 1.32 (3H, d, *J* = 6.6 Hz, H-γ′ Ile), 1.23–1.20 (1H, m, H-β Leu), 0.95–0.89 (15H, m, H-δ, γ′ Ile; H-δ Leu; C-γ Val 2x); $^{13}$C-NMR (150 MHz, CD$_3$OD): $\delta_C$ (ppm) 173.9 (COOH Ile), 172.6 (C=O), 171.5 (C=O), 170.1 (C=O), 62.8 (C-α Pro), 62.1 (C-α Val), 58.6 (C-α Ile), 54.4 (C-α Leu), 48.5 (C-δ Pro), 42.4 (C-β Leu), 37.5 (C-β Ile), 35.6 (C-β Val), 30.3 (C-β Pro), 25.3 (C-γ Ile), 24.9 (C-γ Leu), 24.4 (C-γ Pro), 22.5 (C-δ Leu 2x), 17.3 (C-γ Val 2x), 15.9 (C-γ′ Ile), 11.7 (C-δ Ile).

**PKAI**: $[\alpha]_D^{28}$ − 51.6 (*c* 0.10, MeOH); HR-TOF-ESI-MS *m/z* [M + H]$^+$ 428.2869 (calculated *m/z* 428.2873 for C$_{20}$H$_{38}$N$_5$O$_5$); $^1$H-NMR (600 MHz, CD$_3$OD): $\delta_H$ (ppm) 8.70 (1H, d, *J* = 8.4 Hz, NH), 8.40 (1H, d, *J* = 8.4 Hz, NH), 7.49 (1H, dd, *J* = 4.8, 8.4 Hz, NH), 4.41 (1H, t, *J* = 7.8 Hz, H-α), 4.34–4.30 (2H, m, H-α), 3.40–3.35 (1H, m, H-α), 3.32–3.27 (2H, m, H-δ Pro), 2.91 (2H, t, *J* = 7.8 Hz, H-ω Lys), 2.42–2.39 (1H, m, H-β Ile), 2.07–1.97 (4H, m, H-β Pro; H-β Lys), 1.91–1.80 (2H, m, H-γ Pro), 1.71–1.64 (4H, m, H-δ Lys; H-γ Ile), 1.49–1.45 (3H, m, H-β Ala), 1.26–1.21 (2H, m, H-γ Lys), 0.93–0.88 (6H, m, C-δ Ile; C-γ′ Ile); $^{13}$C-NMR (150 MHz, CD$_3$OD): $\delta_C$ (ppm) 173.3 (COOH Ile), 172.3 (C=O, 2x), 168.4 (C=O), 59.6 (C-α Pro), 56.8 (C-α Ile; C-α Lys), 53.5 (C-α Ala), 47.2 (C-α Ala), 46.1 (C-δ Pro), 39.1 (C-ω Lys), 36.9 (C-β Ile), 31.1 (C-β Lys), 29.7 (C-β Pro), 26.8 (C-δ Lys), 24.8 (C-γ Ile), 23.6 (C-γ Pro), 22.3 (C-γ Lys), 14.7 (C-β Ala), 13.6 (C-γ′ Ile), 10.5 (C-δ Ile).

**PKFI**: $[\alpha]_D^{26}$ + 19.9 (c 0.20, MeOH); HR-TOF-ESI-MS m/z [M + H]$^+$ 504.3178 (calculated m/z 504.3186 for $C_{26}H_{42}N_5O_5$); $^1$H-NMR (500 MHz, CD$_3$OD): $\delta_H$ (ppm) 8.71 (1H, d, J = 5.0 Hz, NH), 8.41 (2H, quint, J = 5.0 Hz, NH), 7.51 (4H, d, J = 5.0 Hz, aryl-H Phe), 7.49–7.48 (1H, m, aryl-H Phe), 4.42 (1H, quint, J = 5.0 Hz, H-α), 4.34–4.30 (3H, m, H-α), 3.39–3.36 (1H, m, H-β Phe), 3.32–3.29 (1H, m, H-β Phe), 2.92–2.89 (1H, m, H-δ Pro), 2.78–2.75 (3H, m, H-δ Pro; H-ω Lys), 2.42–2.39 (1H, m, H-β Ile), 2.08–2.06 (2H, m, H-β Pro), 2.05–1.98 (2H, m, H-β Lys), 1.84–1.82 (2H, m, H-γ Pro), 1.71–1.64 (4H, m, H-γ Ile; H-δ Lys), 1.51–1.45 (2H, m, H-γ Lys), 1.03–0.89 (6H, m, C-δ Ile; C-γ' Ile); $^{13}$C-NMR (150 MHz, CD$_3$OD): $\delta_C$ (ppm) 174.7 (COOH Ile), 172.3 (C=O), 171.0 (C=O), 170.1 (C=O), 135.0 (aryl Cq), 128.6 (aryl CH meta), 127.1 (aryl CH ortho), 120.8 (aryl CH para), 61.9 (C-α Pro), 59.3 (C-α Ile), 58.5 (C-α Phe), 56.8 (C-α Lys), 45.6 (C-δ Pro), 42.0 (C-ω Lys), 38.8 (C-β Ile), 37.3 (C-β Phe), 31.5 (C-β Lys), 31.1(C-β Pro), 28.6 (C-δ Lys), 25.2 (C-γ Ile), 24.8 (C-γ Pro), 22.3 (C-γ Lys), 14.7 (C-γ' Ile), 10.5 (C-δ Ile).

**c-PLPI**: $[\alpha]_D^{28}$ − 94.8 (c 0.10, MeOH); HR-TOF-ESI-MS m/z [M + Na] 443.2641 (calculated m/z 443.2634 for $C_{22}H_{36}N_4O_4Na$); $^1$H-NMR (600 MHz, CDCl$_3$): $\delta_H$ (ppm) 4.64 (1H, dd, J = 8.4, 4.2 Hz, H-α), 4.52 (1H, q, J = 6.0 Hz, H-α), 4.32 (1H, d, J = 7.5 Hz, H-α), 4.29 (1H, m, H-α), 3.83–3.79 (1H, m, H-δ Pro), 3.65–3.61 (1H, m, H-δ Pro), 2.45–2.36 (1H, m, H-β Ile) 2.20–2.14 (1H, m, H-β Pro), 2.09–2.05 (2H, m, H-β Pro), 2.04–2.01 (4H, m, H-β Pro; H-γ Pro), 1.92–1.86 (1H, m, H-γ Pro), 1.79–1.72 (1H, m, H-β Leu), 1.63–1.61 (1H, m, H-β Leu), 1.57–1.49 (1H, m, H-γ Ile), 1.28–1.23 (1H, m, 1H, m, H-γ Ile), 1.09–0.88 (12H, m, H-δ, γ' Ile; H-δ, δ' Leu); $^{13}$C-NMR (150 MHz, CDCl$_3$): $\delta_C$ (ppm) 174.6 (C=O Pro), 174.1 (C=O Pro), 172.9 (C=O Leu), 172.1 (C=O Ile), 60.9 (C-α Ile), 60.5 (C-α Pro), 58.0 (C-α Pro), 51.6 (C-δ Pro), 47.5 (C-δ Pro), 40.6 (C-β Ile), 38.1, 30.6 (C-β Pro), 29.9 (C-β Pro), 25.9 (C-γ Pro), 25.7 (C-γ Pro), 24.6 (C-γ Ile), 23.4 (C-γ Leu), 21.3 (C-δ Leu), 21.0 (C-δ Leu), 15.8 (C-γ' Ile), 11.6 (C-δ Ile).

**c-PLFI**: $[\alpha]_D^{28}$ − 45.3 (c 0.05, MeOH); HR-TOF-ESI-MS m/z [M + H]$^+$ 471.2978 (calculated m/z 471.2971 for $C_{26}H_{39}N_4O_4$); $^1$H-NMR (500 MHz, CDCl$_3$): $\delta_H$ (ppm) 7.25 (4H, d, J = 5.0 Hz, aryl-H Phe), 7.21–7.17 (1H, m, aryl-H Phe), 4.57 (2H, quint, J = 7.5 Hz, H-α), 4.48 (1H, d, J = 7.5 Hz, H-α) 4.41 (1H, t, J = 7.5 Hz, H-α), 3.38–3.32 (1H, m, H-δ Pro), 3.12 (1H, d, J = 6.0 Hz, H-β Phe), 2.87 (1H, t, J = 6.0 Hz, H-β Phe), 2.38–2.31 (1H, m, H-β Ile), 2.04–1.82 (4H, m, H-β Pro; H-γ Pro), 1.64–1.57 (1H, m, H-β Leu), 1.51 (3H, t, J = 6.0 Hz, H-β Leu; H-γ Ile), 1.26–1.18 (1H, m, H-γ Leu), 0.96–0.89 (12H, m, H-δ, γ' Ile; H-δ, δ' Leu); $^{13}$C-NMR (125 MHz, CD$_3$OD): $\delta_C$ (ppm) 173.4 (C=O), 173.3 (C=O), 172.7 (C=O), 168.4 (C=O), 150.8 (aryl Cq), 128.5 (aryl, CH meta), 120.7 (aryl, CH ortho), 119.2 (aryl CH, para), 59.6 (C-α, 2x), 56.8 (C-α), 52.1 (C-α), 48.1 (C-δ Pro), 46.1 (C-β Leu), 40.6 (C-β Ile), 37.1 (C-β Phe), 29.7 (C-β Pro), 24.8 C-γ Pro), 24.5 (C-γ Ile), 23.7 (C-γ Leu), 22.1 (C-δ Leu), 20.4 (C-δ Leu), 16.6, 14.7 (C-γ' Ile), 10.5 (C-δ Ile).

**c-PLVI**: $[\alpha]_D^{28}$ − 47.1 (c 0.10, MeOH); HR-TOF-ESI-MS m/z [M + H]$^+$ 423.2976 (calculated m/z 423.2971 for $C_{22}H_{39}N_4O_4$); $^1$H-NMR (500 MHz, CDCl$_3$): $\delta_H$ (ppm) 8.69 (1H, d, J = 5.0 Hz, NH), 8.39 (1H, d, J = 5.0 Hz, NH), 7.55 (1H, s, NH), 4.52–4.49 (2H, m, H-α), 4.43–4.39 (1H, m, H-α), 4.37 (1H, t, J = 5.0 Hz, H-α), 3.38–3.37 (1H, m, H-δ Pro), 3.35–3.29 (1H, m, H-δ Pro), 2.41–2.40 (1H, m, H-β Val), 2.10–1.99 (4H, m, H-β Ile; H-β Pro; H-γ Pro), 1.87 (1H, t, J = 5.0 Hz, H-γ Pro), 1.68–1.67 (1H, m, H-β Leu), 1.60–1.58 (2H, m, H-γ Ile), 1.48–1.47 (1H, m, H-γ Leu), 1.33–1.33 (3H, m, H-γ' Ile), 1.23–1.21 (1H, m, H-β Leu), 0.95–0.9 (15H, m, H-δ, γ' Ile; H-δ Leu; C-γ Val 2x); $^{13}$C-NMR (125 MHz, CDCl$_3$): $\delta_C$ (ppm) 172.6 (C=O), 171.4 (C=O), 170.1 (C=O), 165.1 (C = O), 62.7 (C-α), 62.1 (C-α), 58.5 (C-α), 54.4 (C-α), 48.4 (C-δ Pro), 42.4 (C-β Leu), 37.5 (C-β Ile), 35.5 (C-β Val), 30.2 (C-β Pro), 25.2 (C-γ Ile), 24.8 (C-γ Leu), 24.4 (C-γ Pro), 22.4 (C-δ Leu 2x), 17.3 (C-γ Val 2x), 15.8 (C-γ' Ile), 11.7 (C-δ Ile).

**c-PKAI**: $[\alpha]_D^{28}$ − 47.9 (c 0.1, MeOH); HR-TOF-ESI-MS m/z [M + H]$^+$ 410.2773 (calculated m.z 410.2767 for $C_{20}H_{36}N_4O_4$); $^1$H-NMR (600 MHz, CDCl$_3$): $\delta_H$ (ppm) 8.66 (1H, d, J = 7.5 Hz, NH), 8.37 (1H, d, J = 7.5 Hz, NH), 7.84 (1H, d, J = 7.5 Hz, NH), 4.43–4.40 (1H, m, H-α), 4.34–4.30 (3H, m, H-α), 3.39–3.37 (1H, m, H-δ Pro), 3.32–3.30 (1H, m, H-δ Pro), 2.89 (2H, t, J = 6.0 Hz, H-ω Lys), 2.42–2.39 (1H, m, H-β Ile), 2.01–1.97 (4H, m, H-β Pro; H-γ Pro), 1.79–1.75 (6H, m, H-β Lys; H-γ Ile; H-δ Lys), 1.50–1.45 (3H, m, H-β Ala), 1.25–1.22 (2H, m, H-γ Lys), 0.98–0.92 (6H, m, H-δ, γ' Ile); $^{13}$C-NMR (150 MHz, CDCl$_3$): $\delta_C$ (ppm) 172.3 (C=O 2x), 168.4 (C=O), 166.2 (C=O), 59.6 (C-α), 56.8 (C-α 2x), 53.5 (C-α), 46.1 (C-δ Pro), 39.1 (C-ω Lys), 37.0 (C-β Ile), 31.1 (C-β Lys), 29.7 (C-β Pro), 26.8 (C-δ Lys), 24.8 (C-γ Ile), 23.6 (C-γ Pro), 22.3 (C-γ Lys), 14.7 (C-γ' Ile), 10.5 (C-δ Ile).

**c-PKFI**: $[\alpha]_D^{26}$ + 16.4 (c 0.10, MeOH); HR-TOF-ESI-MS m/z [M + H]$^+$ 486.2842 (calculated m/z 486.2901 for $C_{26}H_{40}N_4O_4$); $^1$H-NMR (500 MHz, CDCl$_3$): $\delta_H$ (ppm) 8.71 (1H, d, J = 5.0 Hz, NH), 8.41 (2H, q, J = 5.0 Hz), 7.50 (4H, d, J = 5.0 Hz, aryl-H Phe), 7.48–7.49 (1H, m, aryl-H Phe), 4.43 (1H, q, J = 5.0 Hz, H-α), 4.30–4.34 (3H, m, H-α), 3.36–3.39 (1H, m, H-δ Pro), 2.90–3.32 (1H, m, H-δ Pro), 2.89–2.92 (1H, m, H-β Phe), 2.75–2.78 (3H, m, H-β Phe; H-ω Lys), 2.39–2.42 (1H, m, H-β Ile), 1.98–2.08 (2H, m, H-β Pro), 1.64–1.68 (4H, m, H-β Lys), 1.45–1.51 (4H, m, H-γ Ile; H-δ Lys), 1.23–1.26 (2H, m, H-γ Lys), 0.89–1.03 (6H, m, C-γ' Ile; C-δ Ile); $^{13}$C-NMR (125 MHz, CDCl$_3$): $\delta_C$ (ppm) 172.3 (C=O), 171.0 (C=O), 170.1 (C=O), 166.0 (C=O), 135.0 (aryl Cq), 128.6 (aryl CH meta), 127.1 (aryl CH ortho), 120.8 (aryl CH

para), 61.9 (C-α Pro), 59.3 (C-α Ile), 58.5 (C-α Phe), 56.8 (C-α Lys), 45.6 (C-δ Pro), 42.0 (C-ω Lys), 38.8 (C-β Ile), 37.3 (C-β Phe), 31.5 (C-β Lys), 31.1(C-β Pro), 28.6 (C-δ Lys), 25.2 (C-γ Ile), 24.8 (C-γ Pro), 22.3 (C-γ Lys), 14.7 (C-γ′ Ile), 10.5 (C-δ Ile).

## 2.4. Antimicrobial evaluation of linear and cyclic tetrapeptides

### 2.4.1. Preparation of bacterial culture

Tools and materials were sterilized in an autoclave for 15 min at 121°C. Bacterial culture was grown in agar media (Mueller Hinton agar) by picking some bacterial colony using an inoculating loop, followed by incubation at 37°C for 24 h. Bacterial culture was then suspended in liquid media (Mueller Hinton Broth). The suspended bacterial culture was then incubated at 37°C for 24 h and standardized equal to 0.5 Mc Farland ($2 \times 10^8$ CFU ml$^{-1}$). The culture was diluted into the concentration of $5 \times 10^5$ CFU ml$^{-1}$.

### 2.4.2. Preparation of fungal culture

Tools and materials were sterilized in an autoclave for 15 min at 121°C. Fungal culture was grown in agar media (Potato Dextrose agar) by picking some fungal colony using an inoculating loop, followed by incubation at 30°C for 24–48 h. Fungal culture was then suspended in liquid media (Potato Dextrose Broth). The suspended fungal culture was then incubated for 24–48 h at 30°C. The suspended fungal culture was then incubated at 30°C for 24 h and standardized equal to 0.5 Mc Farland ($2 \times 10^8$ CFU ml$^{-1}$). The culture was diluted into the concentration of $5 \times 10^5$ CFU ml$^{-1}$.

### 2.4.3. MIC determination of the synthetic peptides

Six tetrapeptides and six cyclotetrapeptides were dissolved in DMSO 2% with a concentration of 1 µg mL$^{-1}$. Each solution was dissolved gradually. Subsequently, 12 sample solutions, ciprofloxacin, nystatin, ketoconazole and DMSO 2% were incubated on a 96-well microplate at 37°C for 18 h. The result was interpreted using a spectrophotometer, with a wavelength of 600 nm. MIC was measured as the percentage of microbial inhibition and cell death.

# 3. Results and discussion

The syntheses of c-PLAI analogues were carried out based on those of c-PLAI [3]. The linear precursors were synthesized using a solid-phase Fmoc-chemistry method on a 2-chlorotrityl resin, while the cyclic products were obtained through cyclization of the linear peptides in the solution phase. The cyclization site always involved a proline residue at the N-terminus, because the proline residue is a β-turn inducer that can facilitate cyclization [6,7,14]. The synthesis of all peptides began with the attachment of isoleucine to the 2-chlorotrityl resin, which resulted in an isoleucine-resin with loading values in the range of 0.61–0.90 mmol g$^{-1}$ (table 1). During the resin cleavage step of the solid-phase synthesis of PKAI and PKFI, the use of TFE/AcOH instead of TFA allowed us to obtain cleaved linear peptides where the side chain of lysine was still protected with a Boc group. However, the corresponding mass spectra showed the presence of both Boc-protected and Boc-unprotected tetrapeptides at the lysine side chain, which eventually resulted in lower percentage yields of PKAI and PKFI. It is apparent that the TFE/AcOH mixture was not the best choice for resin cleavage, since the Boc protecting group at the side chain of the lysine residue was partially cleaved. During the coupling reactions in the presence of the HATU/HOAt reagent, it was found that triple coupling protocols were needed for the effective coupling between valine and isoleucine as well as between leucine and valine. The presence of the β-branched residue of valine could represent an issue in the synthesis [6,7]. The synthesis of all linear peptides resulted in percentage yields in the range of 29.9–60.8% (table 1). As an example, the synthetic scheme for the preparation of c-PKAI is shown in scheme 1.

All crude linear peptides were lyophilized from acetonitrile/water and analysed using analytical RP-HPLC. The purification step was carried out on a preparative RP-HPLC column to obtain pure linear tetrapeptides. Mass spectrometry as well as $^1$H- and $^{13}$C-NMR spectroscopy experiments were employed to characterize all the tetrapeptides.

The cyclization of the linear tetrapeptides was carried out in dichloromethane ($4 \times 10^{-4}$ M) using HATU and DIPEA as coupling reagent and base, respectively, according to a protocol previously reported[3].

**Scheme 1.** Synthetic route towards c-PKAI. (a) (1) Fmoc-Ile-OH, DIPEA, dichloromethane, 4 h, (2) dichloromethane/MeOH/DIPEA (8 : 1.5 : 0.5), (3) 20% piperidine in DMF, (b) Fmoc-Ala-OH, HOAt, HATU, DIPEA, dichloromethane : DMF (1 : 1), 4 h, (2) 20% piperidine in DMF, (c) (1)Fmoc-Lys(Boc)-OH, HOAt, HATU, DIPEA, dichloromethane : DMF (1 : 1), 4 h, (2) 20% piperidine in DMF, (d) Fmoc-Pro-OH, HOAt, HATU, DIPEA, dichloromethane : DMF (1 : 1), 4 h, (2) 20% piperidine in DMF, (e) AcOH : TFE : DCM (2 : 2 : 6), 1 h, (f) DMF/dichloromethane (0.2 : 99.8), HATU, DIPEA, at 0°C for 1 h, then at 25°C for 30 min, (g) 95% TFA in water.

**Table 1.** Synthesis data for the preparation of the linear precursors.

| compound | resin loading (mmol g$^{-1}$) | crude mass (mg) | pure mass (mg) | yield (%) |
|----------|-------------------------------|-----------------|----------------|-----------|
| PLAI | 0.61 | 192.8 | 171.1 | 61 |
| PLPI | 0.73 | 202.7 | 196.7 | 62 |
| PLVI | 0.90 | 187.4 | 177.9 | 45 |
| PLFI | 0.85 | 250.7 | 229.9 | 55 |
| PKAI | 0.80 | 175.5 | 102.8 | 30 |
| PKFI | 0.82 | 200.1 | 231.8 | 39 |

The reaction was performed at 0°C for 1 h, followed by stirring at 25°C for 30 min. The crude peptides were dissolved in CHCl$_3$ and then washed using HCl, NaHCO$_3$ and NaCl before drying over Na$_2$SO$_4$. After filtration, the filtrate was concentrated to yield crude cyclic peptides. While c-PLVI was sufficiently pure

**Table 2.** Yield data for the cyclization of the linear precursors.

| cyclotetrapeptides | mass (mg) | yield (%) |
|---|---|---|
| c-PLAI | 21 | 70 |
| c-PLPI | 37.4 | 53 |
| c-PLVI | 31.1 | 52 |
| c-PLFI | 64.7 | 46 |
| c-PKAI | 34.2 | 43 |
| c-PKFI | 49.6 | 50 |

**Table 3.** Antimicrobial activities of PLAI and c-PLAI analogues.

| compound | minimum inhibitory concentration (MIC)[a] ($\mu$g ml$^{-1}$) | | | | | |
|---|---|---|---|---|---|---|
| | E. coli ATCC 11229 | K. pneumonia ATCC 2357 | S. aureus ATCC 6538 | B. cereus ATCC 11778 | C. albicans ATCC 10231 | T. mentagrophyte |
| PLAI | >1000 | 250 | >1000 | — | >1000 | — |
| PLPI | 500 | 500 | 500 | 250 | >1000 | 500 |
| PLVI | 500 | 500 | >1000 | 500 | >1000 | — |
| PLFI | >1000 | >1000 | >1000 | 500 | >1000 | — |
| PKAI | 250 | 500 | 500 | 500 | 250 | 500 |
| PKFI | 250 | 500 | 250 | 250 | 250 | 500 |
| c-PLAI | 250 | 500 | >1000 | 125 | >1000 | 125 |
| c-PLPI | 125 | 250 | 250 | — | >1000 | — |
| c-PLVI | 125 | 250 | 500 | 125 | 500 | 62.50 |
| c-PLFI | 125 | 250 | 500 | 125 | 500 | 250 |
| c-PKAI | 62.50 | 250 | 125 | 125 | 125 | 125 |
| c-PKFI | 31.25 | 250 | 62.5 | 62.25 | 62.50 | 125 |
| Ciprofloxacin[b] | 1.56 | 0.391 | 0.098 | 0.195 | — | — |
| Nystatin[b] | — | — | — | — | 4.88 | — |
| Ketokonazole[b] | — | — | — | — | — | 312.5 |

[a]The bioassay was carried out in a single experiment.
[b]Ciprofloxacin was purchased from Fluka Analytical; Nystatin was purchased from an Indonesian pharmaceutical industry, Metiska Farma, with commercial name Kandistatin; Ketokonazole was purchased from Indonesian pharmaceutical industry, Hexpharm Jaya.

and did not require further purification, the other peptides were purified by silica-gel column chromatography using hexane/ethyl acetate as eluent, to obtain the pure c-PLAI analogues in moderate yields (table 2). The characterization of these cyclic products was performed by using mass spectrometry and $^1$H- and $^{13}$C-NMR spectroscopy.

The antimicrobial properties of all linear and cyclic tetrapeptides were evaluated (table 3). The results showed that the parent peptide, i.e. the synthetic c-PLAI, was inactive or weakly active, in contrast with the previous report, explaining that c-PLAI was highly active (MIC values of 6 $\mu$g ml$^{-1}$) against Gram-negative bacteria (*P. aeruginosa* and *K. pneumoniae*) and fungi (*C. albicans* and *T. mentagrophytes*) [2]. The lack of information regarding the microbial strains used in the previous report led us to the assumption that the use of different strains in the assay could lead to a difference in the bioactivity results. It was described that different experimental conditions could also be another reason for the difference observed in the biological testing results [15]. It has been mentioned in several reports that synthetic proline-containing cyclopeptides could possess different biological properties despite being chemically similar [16–18]. The reports described that the conformational change in the proline unit

was indicated as the cause of the contrasting results in the biological activity between the natural and the synthetic materials.

In this study, most of the cyclic peptides exhibited better antimicrobial activities than their linear counterparts against *E. coli*, *K. pneumoniae*, *S. aureus*, *C. albicans* and *T. mentagrophytes*. It seems likely that the cyclic structures could enhance the antimicrobial properties of the peptides. Studies on molecular dynamics showed that a cyclic peptide could bind more strongly to the membrane and penetrate deeper into the bilayer than a linear precursor, thus endowing the cyclic peptide with better antibacterial properties [19]. Cyclic peptides are also known to be more resistant to proteases [20]. In addition, amino acid residues play an important role in antimicrobial activity. The replacement of the alanine residue of c-PLAI with proline, valine, or phenylalanine increased the antimicrobial properties against *E. coli*, *K. pneumoniae* and *S. aureus*. The insertion of lysine to replace the leucine residue of c-PLAI and c-PLFI increased the antimicrobial activity against *E. coli*, *S. aureus* and *C. albicans* both in linear and cyclic peptides. Cationic residues have long been known as common residues found in AMPs [9]. Thus, it is not surprising that c-PKAI and c-PKFI displayed the best antimicrobial properties against almost all strains. Particularly, c-PKFI was active towards *E. coli*, *K. pneumoniae*, *S. aureus*, *B. cereus*, *C. albicans* and *T. mentagrophytes* with MIC values of 31.25, 250, 62.5, 62.5, 62.5 and 125 μg ml$^{-1}$, respectively. The presence of the aromatic residue phenylalanine in c-PKFI also increased the antimicrobial activity compared to c-PKAI that contains alanine residues in its structure. It can be assumed that a combination of cationic and aromatic residues most likely increased the antimicrobial properties of the peptides.

# 4. Conclusion

Cyclo PLAI analogues were successfully synthesized through a combination of solid- and solution-phase methods. Both synthetic linear and cyclic peptides were evaluated for their antimicrobial properties against *E. coli*, *K. pneumoniae*, *S. aureus*, *B. cereus*, *C. albicans* and *T. mentagrophytes*. In our studies, it was found that the synthetic c-PLAI was inactive or weakly active against the tested bacteria. Furthermore, the biological assay results showed that the cyclic peptides of PLAI and its analogues were more biologically active than the corresponding linear structures.

Data accessibility. All the data in this investigation have been reported in the paper and the datasets supporting this article have been uploaded as part of the electronic supplementary material.

Authors' contributions. R.M. and O.I.N. designed the study. R.M. and K.F. wrote the manuscript. O.I.N. synthesized all compounds and evaluated the biological activity of all of said compounds. R.M., O.I.N., J.M.D., A.T.H. and D.S. participated in the data analysis. D.H., N. and U.S. participated in the purification of all of the compounds. All authors gave final approval for publication.

Competing interests. The authors declare no competing interests.

Funding. R.M. would like to thank DIKTI, Unpad and Kemendikbud Indonesia for financial support through the PD grant no. 2925/UN6.D/LT/2019, the ALG grant no. 1427/UN6.3.1/LT/2020 and the WCP-Scheme B grant no. 101.24/E4.3/KU/2020.

Acknowledgements. R.M. would like to thank DIKTI, Unpad and Kemendikbud Indonesia for financial support through the PD grant no. 2925/UN6.D/LT/2019, the ALG grant no. 1427/UN6.3.1/LT/2020 and the WCP-Scheme B grant no. 101.24/E4.3/KU/2020. R.M. would also like to thank Prof. Yoshihito Shiono (Yamagata University) and Dr Mohamad Nurul Azmi (University Sains Malaysia) for NMR measurement of several peptides.

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
