## [Peer Review File · Royal Society Open Science]

Review History

RSOS-191980.R0 (Original submission)

Review form: Reviewer 1

Is the manuscript scientifically sound in its present form?

No

Are the interpretations and conclusions justified by the results?

No

Is the language acceptable?

No

Do you have any ethical concerns with this paper?

No

Have you any concerns about statistical analyses in this paper?

No

Recommendation?

Major revision is needed (please make suggestions in comments)

Comments to the Author(s)

This paper describes the synthesis of some cyclic peptides designed as antimicrobial agents. The chemistry is adequately described and sufficient characterisation data are provided. The scientific completeness therefore depends upon the adequacy of the biological evaluation. The data presented are not sufficient to allow conclusions to be drawn about antimicrobial activity for two principal reasons:

1. Insufficient species have been tested.
2. There are no positive controls in the data presented. It is therefore not possible to have confidence in the numbers quoted. Indeed the authors allude to this possibility in their text. There is also a discrepancy between the properties of one of the authors' compounds with a compound made elsewhere; this requires further investigation and clarification.

If the above two points are acted upon an acceptable paper might be possible to write.

Review form: Reviewer 2**Is the manuscript scientifically sound in its present form?**

Yes

Are the interpretations and conclusions justified by the results?

Yes

Is the language acceptable?

No

Do you have any ethical concerns with this paper?

No

Have you any concerns about statistical analyses in this paper?

No

Recommendation?

Major revision is needed (please make suggestions in comments)

Comments to the Author(s)

Major comments: Improve English throughout the manuscript. Overall, the paper is too simple and appears too preliminary at its present stage. I miss a figure describing all schematics including the design strategy developed for this work. The discussion must be improved (e.g., "It is so unfortunate that the MIC values of linear and cyclic-PLAI against *Staphylococcus aureus* and *Candida albicans* did not close to the result reported by Dahiya and Gautam (2011)."). Such comments are not acceptable for publication.

Major problem: The templates are not active and the design strategy is solely based on the activity of the templates.

Specific Comments:

- Abstract: "Antimicrobial peptides (AMPs) have become interesting compounds due to their ability to kill many pathogens without causing resistance." This is not known to be the case 100%. Resistance assays would be needed to demonstrate this.

- More background on AMPs is needed in the introduction section. Also describe the importance of natural templates and why analogs are needed (e.g., doi: 10.1016/j.jmb.2018.12.015).
- The authors should show resistance to degradation and toxicity assays towards mammalian cells.
- More background needed on why Leu and Ala were selected to substitute aromatic and cationic aa residues.
- Explain the importance of Pro for cyclization (without it the yields that are low would be even lower)
- Synthesis and characterization are good. Using Oxyme resin would have likely given better yields and Fmoc-N-terminal could have been used instead of Boc. Deprotecting the side chains makes sense. "In the solid-phase synthesis of PKAI and PKFI, the cleaved linear peptides were obtained with Boc-protected lysine residue in both structures. TFE/AcOH was employed instead of TFA in the resin cleavage step of both PKAI and PKFI. The lower percentage of yields in the synthesis of PKAI and PKFI seems likely to be caused by the presence of the Boc-protected lysine in the sequences, requiring the resin cleavage using TFE/AcOH." Here, the authors must clarify if they are talking about orthogonal protection and if that is the case, why didn't they use Boc-Lys(Bz)-OH or Boc-Lys(OtBu)-OH instead?
- "The MIC values difference could be resulted from the difference in the conformation of peptide that was found in some proline-containing cyclic peptides as mentioned by Shaheen et al. (2012)." What is meant here? More details and discussion of the results are needed comparing with the literature and other small cyclic peptides.
- Standardize the experiments (same range of concentrations) and please add $\mu\text{mol L}^{-1}$ instead of $\mu\text{g/mL}$, it is no possible to make any comparisons when using $\mu\text{g/mL}$ especially for molecules of that size.
- The conclusion section should be re-written.

Review form: Reviewer 3

Is the manuscript scientifically sound in its present form?

No

Are the interpretations and conclusions justified by the results?

Yes

Is the language acceptable?

Yes

Do you have any ethical concerns with this paper?

No

Have you any concerns about statistical analyses in this paper?

No

Recommendation?

Major revision is needed (please make suggestions in comments)

Comments to the Author(s)

The manuscript entitled "Synthesis of Cyclotetrapeptide Analogues of c-PLAI and Evaluation for Their Antimicrobial Properties" by Maharani et al describes the synthesis of 5 new cyclic peptides and their evaluation as antimicrobial peptides against *E. coli*, *S. aureus*, and *C. albicans*. The cyclic peptides are based on the known natural cyclic peptide c-PLAI. The new cyclic peptides represents an addition to the literature, as does their synthesis.

The experimental methods have significant flaws.

-There are substantial differences in chemical shift between the published cPLAI 1H and 13C NMR from ref 4 and those reported in this study (for example all 4 alpha protons differ by approximately 0.3 ppm chemical shift). This must be addressed.

-Significant information is missing from the methods.

1. The lysine was used in the peptide synthesis is not listed with the other amino acids.
2. Column sizes and mobile phase gradients are not included for HPLC analysis and purification.
3. Exact mass of PLFI, cPLPI, and cPKAI do not match calculated mass within 3mmu. These data thus does not verify the molecular formula therefore the compounds are not fully characterized.
4. Media and strains (eg ATCC number or other specific strain designator) used in the antimicrobial activity assays were not indicated. Details of the seed cultures used are also missing.

This manuscript is not acceptable in its current format.

Decision letter (RSOS-191980.R0)

24-Feb-2020

Dear Dr Maharani:

Manuscript ID: RSOS-191980

Title: "Synthesis of Cyclotetrapeptide Analogues of c-PLAI and Evaluation for Their Antimicrobial Properties"

Thank you for submitting the above manuscript to Royal Society Open Science. Your paper was sent to reviewers and their comments are included at the bottom of this letter.

In view of the concerns raised by the reviewers, the manuscript has been rejected in its current form. However, a new manuscript may be submitted which takes into consideration these comments.

Please note that resubmitting your manuscript does not guarantee eventual acceptance, and that your resubmission will be subject to peer review before a decision is made.

Your resubmitted manuscript should be submitted by 23-Aug-2020. If you are unable to submit by this date please contact the Editorial Office.

On behalf of the Subject Editor Professor Anthony Stace and the Associate Editor Dr Andrew Harned

REVIEWER(S) REPORTS:

Associate Editor Comments to Author ():

RSC Associate Editor:

Comments to the Author:

The reviewers have raised a number of serious concerns regarding the synthesis and biological evaluation of the reported cyclic peptides. Some of the more troubling concerns are, but not limited to, (i) the characterization of the compounds, (ii) the lack of spectral agreement with literature values, (iii) the conformation of the final product, (iv) lack of positive controls during antimicrobial activity assays, and (v) evaluation against a small number of microbial species.

In my opinion, the number of concerns and the amount of work it may take to address them falls outside the threshold for "Major Revisions". But, I encourage the authors to carefully consider the reviewer comments and submit a revised manuscript at a later time.

RSC Subject Editor:

Comments to the Author:

(There are no comments.)

Reviewers' Comments to Author:

Reviewer: 1

Comments to the Author(s)

This paper describes the synthesis of some cyclic peptides designed as antimicrobial agents. The chemistry is adequately described and sufficient characterisation data are provided. The scientific completeness therefore depends upon the adequacy of the biological evaluation. The data presented are not sufficient to allow conclusions to be drawn about antimicrobial activity for two principal reasons:

1. Insufficient species have been tested.
 2. There are no positive controls in the data presented. It is therefore not possible to have confidence in the numbers quoted. Indeed the authors allude to this possibility in their text. There is also a discrepancy between the properties of one of the authors' compounds with a compound made elsewhere; this requires further investigation and clarification.
- If the above two points are acted upon an acceptable paper might be possible to write.

Reviewer: 2

Comments to the Author(s)

Major comments: Improve English throughout the manuscript. Overall, the paper is too simple and appears too preliminary at its present stage. I miss a figure describing all schematics including the design strategy developed for this work. The discussion must be improved (e.g., "It is so unfortunate that the MIC values of linear and cyclic-PLAI against *Staphylococcus aureus* and *Candida albicans* did not close to the result reported by Dahiya and Gautam (2011)."). Such comments are not acceptable for publication.

Major problem: The templates are not active and the design strategy is solely based on the activity of the templates.

Specific Comments:

- Abstract: "Antimicrobial peptides (AMPs) have become interesting compounds due to their ability to kill many pathogens without causing resistance." This is not known to be the case 100%. Resistance assays would be needed to demonstrate this.
- More background on AMPs is needed in the introduction section. Also describe the importance of natural templates and why analogs are needed (e.g., doi: 10.1016/j.jmb.2018.12.015).
- The authors should show resistance to degradation and toxicity assays towards mammalian cells.
- More background needed on why Leu and Ala were selected to substitute aromatic and cationic aa residues.
- Explain the importance of Pro for cyclization (without it the yields that are low would be even lower)
- Synthesis and characterization are good. Using Oxyme resin would have likely given better yields and Fmoc-N-terminal could have been used instead of Boc. Deprotecting the side chains makes sense. "In the solid-phase synthesis of PKAI and PKFI, the cleaved linear peptides were obtained with Boc-protected lysine residue in both structures. TFE/AcOH was employed instead of TFA in the resin cleavage step of both PKAI and PKFI. The lower percentage of yields in the synthesis of PKAI and PKFI seems likely to be caused by the presence of the Boc-protected lysine in the sequences, requiring the resin cleavage using TFE/AcOH." Here, the authors must clarify if they are talking about orthogonal protection and if that is the case, why didn't they use Boc-Lys(Bz)-OH or Boc-Lys(OtBu)-OH instead?
- "The MIC values difference could be resulted from the difference in the conformation of peptide that was found in some proline-containing cyclic peptides as mentioned by Shaheen et al. (2012)." What is meant here? More details and discussion of the results are needed comparing with the literature and other small cyclic peptides.

- Standardize the experiments (same range of concentrations) and please add $\mu\text{mol L}^{-1}$ instead of $\mu\text{g/mL}$, it is no possible to make any comparisons when using $\mu\text{g/mL}$ especially for molecules of that size.
- The conclusion section should be re-written.

Reviewer: 3

Comments to the Author(s)

The manuscript entitled "Synthesis of Cyclotetrapeptide Analogues of c-PLAI and Evaluation for Their Antimicrobial Properties" by Maharani et al describes the synthesis of 5 new cyclic peptides and their evaluation as antimicrobial peptides against *E. coli*, *S. aureus*, and *C. albicans*. The cyclic peptides are based on the known natural cyclic peptide c-PLAI. The new cyclic peptides represents an addition to the literature, as does their synthesis. The experimental methods have significant flaws.

-There are substantial differences in chemical shift between the published cPLAI ^1H and ^{13}C NMR from ref 4 and those reported in this study (for example all 4 alpha protons differ by approximately 0.3 ppm chemical shift). This must be addressed.

-Significant information is missing from the methods.

1. The lysine was used in the peptide synthesis is not listed with the other amino acids.
2. Column sizes and mobile phase gradients are not included for HPLC analysis and purification.
3. Exact mass of PLFI, cPLPI, and cPKAI do not match calculated mass within 3mmu. These data thus does not verify the molecular formula therefore the compounds are not fully characterized.
4. Media and strains (eg ATCC number or other specific strain designator) used in the antimicrobial activity assays were not indicated. Details of the seed cultures used are also missing.

This manuscript is not acceptable in its current format.

Author's Response to Decision Letter for (RSOS-191980.R0)

See Appendix A.

RSOS-201822.R0

Review form: Reviewer 1

Is the manuscript scientifically sound in its present form?

Yes

Are the interpretations and conclusions justified by the results?

Yes

Is the language acceptable?

Yes

Do you have any ethical concerns with this paper?

No

Have you any concerns about statistical analyses in this paper?

No

Recommendation?

Accept with minor revision (please list in comments)

Comments to the Author(s)

The authors have improved this article substantially in line with recommendations from previous reviewers. The antibiotic activity of the compounds prepared is nevertheless on the edge of scientific significance and certainly not of clinical significance in my opinion. This being the case, I think the authors should remove the final sentence of the conclusions. The results do not justify the claim for a broad spectrum antibiotic.

Review form: Reviewer 4

Is the manuscript scientifically sound in its present form?

No

Are the interpretations and conclusions justified by the results?

No

Is the language acceptable?

No

Do you have any ethical concerns with this paper?

No

Have you any concerns about statistical analyses in this paper?

No

Recommendation?

Reject

Comments to the Author(s)

Reference: RSOS-201822

Synthesis of Cyclotetrapeptide Analogs of c-PLAI and Evaluation of Their Antimicrobial Properties

Review comments

This manuscript describes the synthesis and antimicrobial (antibacterial and antifungal) activities of a small set of cyclic tetrapeptides.

However, there are a number of issues that would need to be addressed before publication could be considered.

General comments

- 1) The number of cyclotetrapeptides is small (6, with 5 new molecules) making it very difficult to extract any meaningful SAR trends, though the results reported are of interest.
- 2) The authors need to put the results of their study in the context of what is known in general for small cyclic peptides, etc. Some references which need to be considered include, for example, i) Chem. Rev .2019 10318 Cyclic Tetrapeptides from Nature and Design; ii) J Antibiot 2014 541 Two new cyclic tetrapeptides from deep-sea bacterium.; iii) J Mol Biol 2019 3547 Peptide Design Principles for Antimicrobial Applications; iv) J Chem Theory Comput 2013 650 Exploring the energy landscape of cyclic tetrapeptides with discrete path sampling; v) Mar Drugs 2015 3029 Antitumor and Antimicrobial Activity of Some Cyclic Tetrapeptides and Tripeptides Derived from Marine Bacteria.
- 3) Check the use of a, an and the throughout the manuscript, particularly in the Experimental Section.

Specific comments

- The authors have not addressed the original reviewer 2's comment regarding the conformation of these cyclic peptides, which is a critical aspect. It is alluded to but nothing concrete is demonstrated or discussed when it is written "In a report by Shaheen et al. (2012), it was mentioned that some studies disclosed that synthetic proline-containing cyclopeptides could possess different biological properties despite being chemically similar. The conformational difference in the cyclopeptides was indicated as the cause of the contrasting results." This really needs to be addressed in detail if publication is to be considered.
- Gram-positive, etc. requires a capital letter throughout as it is a person's name
- The description of the design strategy for the chosen peptides needs to be expanded. Why was the Ile residue kept throughout – was there a reason for this? Comment on the relative positioning of the Lys, Phe and Val residues in the targeted peptides – see references mentioned in point 2 of general comments above.
- Need to give detail of the method (and the literature reference) used for the measurement of the resin loading using absorbance measurements. There are a number of different reported literature methods which use different wavelengths for measurement and extinction coefficients of the piperidine-DBF adduct.
- In the experimental description it would be better to describe the "resin as being filtered" rather than dried. Drying implies the removal of all traces of solvents, to dryness.
- Give details of the exact TLC solvent system used in the cyclisation step, as well as the type of TLC plates used and silica gel used for column chromatography.
- In the experimental procedure there is confusion over the use of the word "extracted". Extraction infers that a useful chemical is removed from the solution. "Washed" would be a better description to use.
- Measurement of the $[\alpha]_D$ values of the 6 cyclic peptides would be very useful for anyone repeating this work.
- Some of the cyclic peptides were purified by silica gel column chromatography using hexane/ethyl acetate (1:1) as the eluent. References 1 and 3 used semi-preparative HPLC for

purification. Is this correct, as reference 3 is by the authors? Reference 2 lists R_f values for peptides (e.g. c-PLAI) in CHCl₃/MeOH (9:1 or 7:3), a significantly more polar eluent system.

- For the PKAI and PKFI peptides it is stated that the Lysine Boc-protected and Boc-unprotected peptides were seen by MS. Were these both visible in the HPLC analyses or was the Boc group lost in the MS analysis?
- As it is likely that the conformation of the peptides is critical, more detailed assignments of the NMR data would be useful, as was done in the authors' previous publication (reference 3). This would allow anyone repeating this work to have more confidence in the data reported, as well as making comparisons more simple. The exact conformations of all of the peptides is critical in order to be able to make any sense of the biological results. These studies are very important.
- It is more usual to list ¹H and ¹³C NMR chemical shifts from downfield to upfield (reading left to right on the spectrum), as was done in the authors' reference 3 publication.
- Need to recheck the multiplicities of some of the ¹H NMR signals listed. For example, in c-PLPI and c-PLVI, there are quartets listed at ~4.6-4.7 ppm. Are these signals quartets or "doublet of doublets" (dd)? If a dd is present, then two J values are required. Check throughout.
- In all cases where d, t or q are listed in the ¹H NMR data then the corresponding J values must be listed in every case.
- For the antimicrobial evaluation, with the current level of detail it would be difficult to repeat this work. More detail is required or else a literature reference which contains the required level of detail, in order to be able to repeat the assay. Some detail on the choice of bacterial and fungal strains used in the study would also be useful.
- Need to state whether the MIC values measured and reported in Table 3 were single measurements, duplicates or triplicates, etc.
- Although the references are listed in numerical order, when they are mentioned in the text they are referred to with the author and year of publication. This is confusing.
- Spelling of chlorotrityl is wrong twice in the manuscript
-"presence of HATU/HOBT" not "present..."
- What was the source of the ciprofloxacin, nystatin and ketoconazole used?
- For Tables 1 and 2 it would be more usual to list % yield values as whole numbers.

In its current state I can not recommend the manuscript for publication.

Decision letter (RSOS-201822.R0)

The editorial office reopened on 4 January 2021. We are working hard to catch up after the festive break. If you need advice or an extension to a deadline, please do not hesitate to let us know -- we

will continue to be as flexible as possible to accommodate the changing COVID situation. We wish you a happy New Year, and hope 2021 proves to be a better year for everyone.

Dear Dr Maharani:

Title: Synthesis of Cyclotetrapeptide Analogs of c-PLAI and Evaluation of Their Antimicrobial Properties

Manuscript ID: RSOS-201822

The editor assigned to your paper has now received comments from reviewers. We would like you to revise your paper in accordance with the referee and Subject Editor suggestions which can be found below (not including confidential reports to the Editor). Please note this decision does not guarantee eventual acceptance.

Please submit a copy of your revised paper before 05-Feb-2021. Please note that the revision deadline will expire at 00.00am on this date. If we do not hear from you within this time then it will be assumed that the paper has been withdrawn. In exceptional circumstances, extensions may be possible if agreed with the Editorial Office in advance. We do not allow multiple rounds of revision so we urge you to make every effort to fully address all of the comments at this stage. If deemed necessary by the Editors, your manuscript will be sent back to one or more of the original reviewers for assessment. If the original reviewers are not available we may invite new reviewers.

On behalf of the Subject Editor Professor Anthony Stace and the Associate Editor Dr Andrew Harned.

RSC Associate Editor

Comments to the Author:

The authors have submitted a revised manuscript that addressed many of the concerns raised in the previous review. I attempted to get guidance from the original referees, but unfortunately only one of them responded. As a result, and given the extensive requests that resulted from the initial review, I felt it was prudent to get guidance from an additional referee. This referee has raised an additional set of concerns that are all valid and would significantly improve the manuscript. In particular, this reviewer has identified a key point from the previous review that was not sufficiently addressed by the authors.

Reviewers' Comments to Author:

Reviewer: 1

Comments to the Author(s)

The authors have improved this article substantially in line with recommendations from previous reviewers. The antibiotic activity of the compounds prepared is nevertheless on the edge of scientific significance and certainly not of clinical significance in my opinion. This being the case, I think the authors should remove the final sentence of the conclusions. The results do not justify the claim for a broad spectrum antibiotic.

Reviewer: 4

Comments to the Author(s)

Reference: RSOS-201822

Synthesis of Cyclotetrapeptide Analogs of c-PLAI and Evaluation of Their Antimicrobial Properties

Review comments

This manuscript describes the synthesis and antimicrobial (antibacterial and antifungal) activities of a small set of cyclic tetrapeptides.

However, there are a number of issues that would need to be addressed before publication could be considered.

General comments

- 1) The number of cyclotetrapeptides is small (6, with 5 new molecules) making it very difficult to extract any meaningful SAR trends, though the results reported are of interest.
- 2) The authors need to put the results of their study in the context of what is known in general for small cyclic peptides, etc. Some references which need to be considered include, for example, i) Chem. Rev. 2019 10318 Cyclic Tetrapeptides from Nature and Design; ii) J Antibiot 2014 541 Two new cyclic tetrapeptides from deep-sea bacterium.; iii) J Mol Biol [2019 3547](tel:2019 3547) Peptide Design Principles for Antimicrobial Applications; iv) J Chem Theory Comput 2013 650 Exploring the energy landscape of cyclic tetrapeptides with discrete path sampling; v) Mar Drugs [2015 3029](tel:2015 3029) Antitumor and Antimicrobial Activity of Some Cyclic Tetrapeptides and Tripeptides Derived from Marine Bacteria.

3) Check the use of a, an and the throughout the manuscript, particularly in the Experimental Section.

Specific comments

- The authors have not addressed the original reviewer 2's comment regarding the conformation of these cyclic peptides, which is a critical aspect. It is alluded to but nothing concrete is demonstrated or discussed when it is written "In a report by Shaheen et al. (2012), it was mentioned that some studies disclosed that synthetic proline-containing cyclopeptides could possess different biological properties despite being chemically similar. The conformational difference in the cyclopeptides was indicated as the cause of the contrasting results." This really needs to be addressed in detail if publication is to be considered.
- Gram-positive, etc. requires a capital letter throughout as it is a person's name
- The description of the design strategy for the chosen peptides needs to be expanded. Why was the Ile residue kept throughout – was there a reason for this? Comment on the relative positioning of the Lys, Phe and Val residues in the targeted peptides – see references mentioned in point 2 of general comments above.
- Need to give detail of the method (and the literature reference) used for the measurement of the resin loading using absorbance measurements. There are a number of different reported literature methods which use different wavelengths for measurement and extinction coefficients of the piperidine-DBF adduct.
- In the experimental description it would be better to describe the "resin as being filtered" rather than dried. Drying implies the removal of all traces of solvents, to dryness.
- Give details of the exact TLC solvent system used in the cyclisation step, as well as the type of TLC plates used and silica gel used for column chromatography.
- In the experimental procedure there is confusion over the use of the word "extracted". Extraction infers that a useful chemical is removed from the solution. "Washed" would be a better description to use.
- Measurement of the $[\alpha]_D$ values of the 6 cyclic peptides would be very useful for anyone repeating this work.
- Some of the cyclic peptides were purified by silica gel column chromatography using hexane/ethyl acetate (1:1) as the eluent. References 1 and 3 used semi-preparative HPLC for purification. Is this correct, as reference 3 is by the authors? Reference 2 lists R_f values for peptides (e.g. c-PLAI) in CHCl₃/MeOH (9:1 or 7:3), a significantly more polar eluent system.
- For the PKAI and PKFI peptides it is stated that the Lysine Boc-protected and Boc-unprotected peptides were seen by MS. Were these both visible in the HPLC analyses or was the Boc group lost in the MS analysis?
- As it is likely that the conformation of the peptides is critical, more detailed assignments of the NMR data would be useful, as was done in the authors' previous publication (reference 3). This would allow anyone repeating this work to have more confidence in the data reported, as well as making comparisons more simple. The exact conformations of all of the peptides is critical in order to be able to make any sense of the biological results. These studies are very important.

- It is more usual to list ^1H and ^{13}C NMR chemical shifts from downfield to upfield (reading left to right on the spectrum), as was done in the authors' reference 3 publication.
- Need to recheck the multiplicities of some of the ^1H NMR signals listed. For example, in c-PLPI and c-PLVI, there are quartets listed at $\sim 4.6\text{--}4.7$ ppm. Are these signals quartets or "doublet of doublets" (dd)? If a dd is present, then two J values are required. Check throughout.
- In all cases where d, t or q are listed in the ^1H NMR data then the corresponding J values must be listed in every case.
- For the antimicrobial evaluation, with the current level of detail it would be difficult to repeat this work. More detail is required or else a literature reference which contains the required level of detail, in order to be able to repeat the assay. Some detail on the choice of bacterial and fungal strains used in the study would also be useful.
- Need to state whether the MIC values measured and reported in Table 3 were single measurements, duplicates or triplicates, etc.
- Although the references are listed in numerical order, when they are mentioned in the text they are referred to with the author and year of publication. This is confusing.
- Spelling of chlorotriptyl is wrong twice in the manuscript
-" presence of HATU/HOBT" not "present..."
- What was the source of the ciprofloxacin, nystatin and ketoconazole used?
- For Tables 1 and 2 it would be more usual to list % yield values as whole numbers.

In its current state I can not recommend the manuscript for publication.

Author's Response to Decision Letter for (RSOS-201822.R0)

See Appendix B.

Decision letter (RSOS-201822.R1)

Dear Dr Maharani:

Title: Synthesis of Cyclotetrapeptide Analogs of c-PLAI and Evaluation of Their Antimicrobial Properties

Manuscript ID: RSOS-201822.R1

It is a pleasure to accept your manuscript in its current form for publication in Royal Society Open Science. The chemistry content of Royal Society Open Science is published in collaboration with the Royal Society of Chemistry.

On behalf of the Subject Editor Professor Anthony Stace and the Associate Editor Dr Andrew Harned.

RSC Associate Editor

Comments to the Author:

I believe the authors have addressed the concerns raised by the referees to the best of their ability. I am of the opinion, that this manuscript is now suitable for publication.

Reviewer(s)' Comments to Author:

Appendix A

Dear Editor,

First of all, I would like to thank the RSOS editorial board to pass my first submitted manuscript to three reviewers that have given a lot of input to improve our manuscript. Also, I would like to give a big thank for the extended time given to me for make the new resubmission.

During August-October, my government gave a research grant allowing us to invite a world class professor (WCP) to help Indonesian researcher improving their skill for making the publication. I proposed a rejected manuscript to be fine-tuned by the WCP and has resulted in the final new version of manuscript I submitted today. Our WCP is Prof. Koichi Fukase from Osaka University that helped me to revise the manuscript and also to improve it until it has been a more qualified manuscript (based on my opinion). For all of the assistance that have been given by Prof. Fukase, our team and I agree to include prof Fukase as new co-author and also an additional corresponding author beside me. The consents from all co-authors (replied emails) was attached below.

Most of the comments from the reviewers have been revised as follows:

No	Comments	Response
REVIEWER 1		
1	Insufficient species have been tested.	Two bacteria and one fungus have been added as bioindicators
2	There are no positive controls in the data presented. It is therefore not possible to have confidence in the numbers quoted.	Positive controls have been included now.
REVIEWER 2		
1	Improve English throughout the manuscript.	The English in the manuscript have been improved by the assistance from Prof. Fukase and in the final step, Prof. Fukase has also sent the manuscript to the English Editing Service with his own expense.
2	The discussion must be improved	It has been improved with the assistance of Prof. Fukase
3	The templates are not active and the design strategy is solely based on the activity of the templates.	The template now has a low antibacterial activity against one of the bioindicators
4	Abstract: "Antimicrobial peptides (AMPs) have become interesting compounds due to their ability to kill many pathogens without causing resistance." This is not known to be the case 100%. Resistance assays would be needed to demonstrate this.	The information about the resistance has been removed from the abstract since we do not work in the resistance assay.
5	More background on AMPs is needed in the introduction section. Also describe the importance of natural templates and why analogs are needed (e.g., doi: 10.1016/j.jmb.2018.12.015).	The introduction section has been expanded.

6	Explain the importance of Pro for cyclization (without it the yields that are low would be even lower)	The explanation has been included in the discussion section
7	The MIC values difference could be resulted from the difference in the conformation of peptide that was found in some proline-containing cyclic peptides as mentioned by Shaheen et al. (2012).” What is meant here? More details and discussion of the results are needed comparing with the literature and other small cyclic peptides.	The detail explanation has been included now
8	The conclusion section should be re-written.	It has been rewritten
REVIEWER 3		
1	The lysine was used in the peptide synthesis is not listed with the other amino acids.	The lysine has been listed now
2	Column sizes and mobile phase gradients are not included for HPLC analysis and purification.	The Column sizes and mobile phase gradients has been included now
3	Exact mass of PLFI, cPLPI, and cPKAI do not match calculated mass within 3mmu. These data thus does not verify the molecular formula therefore the compounds are not fully characterized.	The measured masses of the peptides now have a close mass with the calculated ones
4	Media and strains (eg ATCC number or other specific strain designator) used in the antimicrobial activity assays were not indicated. Details of the seed cultures used are also missing.	Media and strains (ATCC) have been included now.

The table above are some revisions we have made for the comments that have been given by the three reviewers. I hope the review process for the revised manuscript will give positive comments.

Regards,

Rani

Consent to include Prof Koichi Fukase as co-author and corresponding author

Inbox x

Rani Maharani <r.maharani@unpad.ac.id>

9:28 AM (1 hour ago)

to Unang, ace, dadan.sumiarsa, Desi, Nurlaelasari, Jelang, Orin ▾

Dear Co-authors,

As the previous manuscript with the title of "Synthesis of Cyclotetrapetides Analogs of c-PLAI and Evaluation of Their Antimicrobial Properties" has been rejected last February by RSOS, The RSOS still gives an opportunity for us to make a new resubmission this October.

Fortunately, we have an opportunity to make some improvements for the manuscript through the WCP-scheme B programme by inviting Prof. Koichi Fukase to give the fine tuning manuscript programme where Prof Fukase has made the article being very good now.

However, I will need the consent from all co-authors to simply reply YES if you agree if Prof. Koichi Fukase can be included as new co-author and also corresponding author. or You can also reply NO if you do not agree with the inclusion.

I am looking forward to hearing from you as soon as possible, as I will submit the manuscript this week.

Regards,

Rani

—

Rani Maharani, Ph.D.
Department of Chemistry
Faculty of Mathematics and Natural Sciences
Universitas Padjadjaran
Jalan Raya Bandung-Sumedang Km 21 45363 Jatinangor
West Java, Indonesia
+62-22-7794391

Desi Harneti

9:48 AM (1 hour ago)

to ace, dadan.sumiarsa, Nurlaelasari, Jelang, Orin, me, Unang ▾

Dear Dr. Rani,

Thank you for your email.

Yes, I agree to include Prof Fukase as new co author and corresponding author.

Ragards,
Desi Harneti

...

DADAN SUMIARSA

10:01 AM (1 hour ago)

to Desi, me, Unang, ace, Nurlaelasari, Jelang, Orin ▾

Yes, I agree.

...

Jelang Muhammad Dirgantara

10:05 AM (1 hour ago)

to me ▾

Dear Dr. Rani Maharani

Thank you for your email

Yes, I agree to include Prof. Fukase as new co-author and corresponding author

Best regards,
Jelang

...

Orin Inggriani Napitupulu

to DADAN, Desi, Unang, ace, Nurlaelasari, me, Jelang ▾

11:02 AM (27 minutes ago)

Dear Ibu Dr. Rani Maharani,

Regarding to the manuscript,
Yes, I totally agree to include Prof. Koichi Fukase as new co-author and corresponding author.
All the best,
Thankyou

Regards,
Orin

Sent from Mail for Windows 10

Unang Supratman

to me ▾

9:39 AM (1 hour ago)

Ya kita tunggu jawaban Prof. Fukase.

salam,unang

Appendix B

RESPONSE TO REVIEWER'S COMMENTS

No	Comments	Response
	REVIEWER 1	
1	Comments to the Author(s) The authors have improved this article substantially in line with recommendations from previous reviewers. The antibiotic activity of the compounds prepared is nevertheless on the edge of scientific significance and certainly not of clinical significance in my opinion. This being the case, I think the authors should remove the final sentence of the conclusions. The results do not justify the claim for a broad spectrum antibiotic.	The final sentence of the conclusion has been removed.
	REVIEWER 4	
	General Comments	
1	The number of cyclotetrapeptides is small (6, with 5 new molecules) making it very difficult to extract any meaningful SAR trends, though the results reported are of interest.	-
2	The authors need to put the results of their study in the context of what is known in general for small cyclic peptides, etc. Some references which need to be considered include, for example, i) Chem. Rev .2019 10318 Cyclic Tetrapeptides from Nature and Design; ii) J Antibiot 2014 541 Two new cyclic tetrapeptides from deep-sea bacterium.; iii) J Mol Biol 2019 3547 Peptide Design Principles for Antimicrobial Applications; iv) J Chem Theory Comput 2013 650 Exploring the energy landscape of cyclic tetrapeptides with discrete path sampling; v) Mar Drugs 2015 3029 Antitumor and Antimicrobial Activity of Some Cyclic Tetrapeptides and Tripeptides Derived from Marine Bacteria.	Four suggested literatures have been cited in the manuscript.
3	Check the use of a, an and the throughout the manuscript,	All of the articles (a, an, the) have been checked by the proofreader.

	particularly in the Experimental Section.	I have attached the screenshots of the corrections, given by the proofreader, below the table.
	Specific comments	
1	The authors have not addressed the original reviewer 2's comment regarding the conformation of these cyclic peptides, which is a critical aspect. It is alluded to but nothing concrete is demonstrated or discussed when it is written "In a report by Shaheen et al. (2012), it was mentioned that some studies disclosed that synthetic proline-containing cyclopeptides could possess different biological properties despite being chemically similar. The conformational difference in the cyclopeptides was indicated as the cause of the contrasting results." This really needs to be addressed in detail if publication is to be considered.	I have put more references to strengthen the explanation about the conformational arrangement that was suspected to be the cause of the biological difference. In addition to this, I also found that the natural product, c-PLAI, isolated by Rungprom et al. showed no antimicrobial activity against some bioindicators that is difference with the bioactivity of the synthetic c-PLAI reported by Dahiya and Gautam. Beside the conformational difference reason, I also have previously mentioned that the difference in the experimental condition can also influence the contrasting results in the biological activity.
2	Gram-positive, etc. requires a capital letter throughout as it is a person's name	gram has been replaced into Gram
3	The description of the design strategy for the chosen peptides needs to be expanded. Why was the Ile residue kept throughout – was there a reason for this? Comment on the relative positioning of the Lys, Phe and Val residues in the targeted peptides – see references mentioned in point 2 of general comments above.	The description of the design strategy for the chosen peptides has been expanded. The reasons why Pro and Ile were kept in the structure have been included in the manuscript. In the strategy, Leu or Ala was substituted with hydrophobic or cationic residues using site-directed mutagenesis approach. The replacement was undertaken to enhance the antimicrobial properties.
4	Need to give detail of the method (and the literature reference) used for the measurement of the resin loading using absorbance measurements. There are a number of different reported literature methods which use different wavelengths for measurement and extinction coefficients of the piperidine-DBF adduct.	The measurement of resin loading used a protocol described in "P. D. White and W. C. Chan, Basic Procedures, in Fmoc Solid Phase Peptide Synthesis, eds. W. C. Chan and P. D. White, Oxford University Press Inc., New York, 2000, p 63." The detail procedure and the references have been included in the experimental section.
5	In the experimental description it would be better to describe the "resin as being filtered" rather than	Dried has been replaced by filtered.

	dried. Drying implies the removal of all traces of solvents, to dryness.	
6	Give details of the exact TLC solvent system used in the cyclisation step, as well as the type of TLC plates used and silica gel used for column chromatography.	TLC solvent system, type of TLC plates and silica gel have been added into the experimental section.
7	In the experimental procedure there is confusion over the use of the word “extracted”. Extraction infers that a useful chemical is removed from the solution. “Washed” would be a better description to use.	Extracted has been changed into washed.
8	Measurement of the $[\alpha]_D$ values of the 6 cyclic peptides would be very useful for anyone repeating this work.	$[\alpha]_D$ values were measured for all samples.
9	Some of the cyclic peptides were purified by silica gel column chromatography using hexane/ethyl acetate (1:1) as the eluent. References 1 and 3 used semi-preparative HPLC for purification. Is this correct, as reference 3 is by the authors? Reference 2 lists R _f values for peptides (e.g. c-PLAI) in CHCl ₃ /MeOH (9:1 or 7:3), a significantly more polar eluent system.	All crude linear peptides and c-PLAI were purified through preparative HPLC. However, all crude cyclic analogues were purified by using SiGel column chromatography (n -hexane:ethyl acetate 1:1).
10	For the PKAI and PKFI peptides it is stated that the Lysine Boc-protected and Boc-unprotected peptides were seen by MS. Were these both visible in the HPLC analyses or was the Boc group lost in the MS analysis?	The HR-TOF-MS spectrum that showed peaks of both lysine Boc-protected and Boc-unprotected was obtained from the crude of the linear peptide. RP-HPLC analysis showed several peaks in the chromatogram that we believe one of the peaks is the lysine Boc-unprotected.
11	As it is likely that the conformation of the peptides is critical, more detailed assignments of the NMR data would be useful, as was done in the authors’ previous publication (reference 3). This would allow anyone repeating this work to have more confidence in the data reported, as well as making comparisons more simple. The exact conformations of all of the peptides is critical in order to be able to make any sense of the	NMR assignments for all analogues have followed the assignment presented in the reference 3.

	biological results. These studies are very important.	
12	It is more usual to list ^1H and ^{13}C NMR chemical shifts from downfield to upfield (reading left to right on the spectrum), as was done in the authors' reference 3 publication.	The list of ^1H and ^{13}C NMR chemical shifts has been revised as suggested.
13	Need to recheck the multiplicities of some of the ^1H -NMR signals listed. For example, in c-PLPI and c-PLVI, there are quartets listed at ~4.6-4.7 ppm. Are these signals quartets or "doublet of doublets" (dd)? If a dd is present, then two J values are required. Check throughout.	All of multiplicities including the J values of all ^1H -NMR signals have been re-checked and the incorrect information have been revised.
14	In all cases where d, t or q are listed in the ^1H -NMR data then the corresponding J values must be listed in every case.	All of J values of all ^1H -NMR signals have been re-checked and the incorrect information have been revised.
15	For the antimicrobial evaluation, with the current level of detail it would be difficult to repeat this work. More detail is required or else a literature reference which contains the required level of detail, in order to be able to repeat the assay. Some detail on the choice of bacterial and fungal strains used in the study would also be useful.	The detail protocol has been described in the experimental section. The information about the strain of bacteria and fungi have been included in the table.
16	Need to state whether the MIC values measured and reported in Table 3 were single measurements, duplicates or triplicates, etc.	The MIC values were measured and reported in a single measurement. The information has been included in the additional information on Table 2.
17	Although the references are listed in numerical order, when they are mentioned in the text they are referred to with the author and year of publication. This is confusing.	The inconsistencies have been revised.
18	Spelling of chlorotriptyl is wrong twice in the manuscript	The incorrect chlorotriptyl spelling have been revised
19"presence of HATU/HOBT" not "present..."	The incorrect word has been revised
20	What was the source of the ciprofloxacin, nystatin and ketoconazole used?	Ciprofloxacin was purchased from Fluka Analytical; Nystatin was purchased from Metiska Farma, an Indonesian pharmaceutical industry with a commercial name of Kandistatin; Ketokonazole was purchased from

		Hexpharm Jaya, an Indonesian pharmaceutical industry. This information has been included in the manuscript
21	For Tables 1 and 2 it would be more usual to list % yield values as whole numbers.	The % yield values have been converted into whole numbers

The manuscript was proofread at <https://qualityproofreading.co.uk>.

Dear Rani,

Please find your edited document here. (Link will expire within 21 days.)

As you will see, all of the editor's changes are tracked so that you can see where the improvements have been made. The most convenient way of removing the tracking notes is to: 1) select the *Review* tab 2) select *Accept Changes* 3) click *Accept All*.

MS Word 2013:

The *Track Changes* highlighting will not show automatically in MS Word 2013. To show changes:

- Select the *Review* tab.
- Click on the dropdown box that currently says *Simple Markup*.
- Select *All Markup* from the options.

The editor commented:

Hi,

I changed US spelling to UK.

I also suggested some alternative words which you can use to avoid repetition.

Finally, sentence structure was altered slightly.

Kind regards,

Michael M

Thank you very much for using **Proofreading Service UK**.

Any questions, please do ask.

Francis Smith
Proofreading Service UK

EXPERIMENTAL SECTION

All the compounds were analysed using the analytical RP-HPLC Waters 2998 Photodiode Array Detector, LiChrospher 100 RP-18 5 mm column. The purification took advantage of Buchi C-620 Sepacore with C18 column (12 g). ^1H - and ^{13}C -NMR spectra were recorded on Agilent NMR 500 MHz (^1H) and 125 MHz (^{13}C) using deuterated solvent, JEOL NMR 600 MHz (^1H) and 150 MHz (^{13}C), and BRUKER NMR 600 MHz (^1H) and 150 MHz (^{13}C). Mass spectrometry spectra were recorded on Waters HR-TOF-MS Lockspray. Loading resin absorbance was measured using the TECAN Infinite pro 200 UV-Vis Spectrophotometer. All of the amino acid residues, Fmoc-L-proline, Fmoc-L-leucine, Fmoc-L-alanine, Fmoc-L-isoleucine, Fmoc-L-phenylalanine, Fmoc-L-valine, Fmoc-L-lysine (Boc), and 2-chlorotrityl chloride resin (0.972 mmol/g), were purchased from GL-Biochem Ltd., China. Dichloromethane, *N,N*-dimethylformamide (DMF), trifluoroethanol, acetic acid, and trifluoroacetic acid used in the synthesis were of analytical grade.

1. Synthesis and Purification of Linear Tetrapeptides

Linear tetrapeptides were synthesised using the manual solid-phase peptide synthesis procedure with 2-chlorotrityl chloride employed as the solid support. 5 mL of dichloromethane

was added into 2-chlorotrityl chloride resin (400 mg, 0.39 mmol) and the resin was shaken with a rotary suspension mixer for 5 minutes. Dichloromethane was removed from the reactor by filtration using an air pump. The first amino acid (AA₁) (0.9 mmol) in 5 mL of dichloromethane and DIPEA (387.2 μL ; 2.22 mmol) was loaded into the resin for 24 hours at room temperature. The resin was dried with an air pump after the resin loading. The loading resin value (amount of amino acid loaded in 1 g of resin) was determined by deprotecting the Fmoc group of the first amino acid attached to 0.5 mg resin by 20% piperidine in DMF for 1 hour. The total amount of Fmoc deprotected from the amino acid was quantified using Spectrophotometer UV-VIS, with a wavelength of 290 nm. The total amount of Fmoc equals the total number of amino acids loaded on the resin. Afterwards, the resin was capped with a 10 mL solution of MeOH:DIPEA:DCM (1.5:0.5:8). After the resin had been capped, the Fmoc group of the first amino acid was deprotected using 20% piperidine in DMF for 30 minutes. Following this, the resin was dried and washed using 3 \times 3 mL DMF and 3 \times 3 mL dichloromethane. The success of the Fmoc deprotection was analysed via a chloranil test. If the test gave the result of green or red resin after 5 minutes of reaction, this indicated that the deprotection was successful, but if the test gave the result of yellow resin, this indicated that the deprotection was unsuccessful and the deprotection needed to be repeated. After Fmoc deprotection, the solution of the second amino acid (AA₂) (2 equiv.), HATU (2 equiv.), HOAt (2 equiv.), and DIPEA (8 equiv.) in 5 mL of dichloromethane:DMF (1:1) was ready to be coupled with the first amino acid on the resin using a rotary suspension mixer for 24 hours. Subsequently, the resin was dried and washed using 3 \times 3 mL DMF and 3 \times 3 mL dichloromethane. The success of the amino acid coupling was analysed via a chloranil test. If the test gave the result of yellow resin after 5 minutes of reaction, this indicated that the coupling was successful, but if the test gave the result of green or blue resin, this indicated that the coupling was unsuccessful and the coupling needed to be repeated. The repetitive deprotection and coupling process was carried out until the desired tetrapeptide attached to the resin. Tetrapeptide on the resin was cleaved using 5 mL of 5% of TFA in dichloromethane for 30 minutes for PLPI, PLVI, and PLFI or 5 mL of AcOH:TfE:DCM (2:2:6) for PKAI and PKFI. The solution was collected and evaporated for further purification. Crudes of PLPI, PLVI, PLFI, PKAI, and PKFI were purified using preparative RP-HPLC with Agilent Pursuit 5 C-18 (250 mm \times 21.2 mm, 5 μL) with eluent of water:acetonitrile (5-95%) for 60 minutes, and flow rate of 3.5-5.0 mL/minute. Each fraction

Michael Deleted: z

Michael Deleted: by

Michael You seem to be leaving a gap between figures and units of measure, so I have made changes here.

Michael Formatted: English (UK)

Michael Formatted: English (UK)

Michael Deleted: on August 12, 2020

Michael Deleted: in

Michael Deleted: z

Michael Deleted: through

Michael Deleted: on

Michael Deleted: A

Michael Deleted: in ... room temperature. The resin was dried with an air pump after the resin loading. The loading resin value (amount of amino acid loaded in 1 g of resin) was determined by deprotecting the Fmoc group of the first amino acid attached on

Michael Is this what you mean?

Michael Deleted: .

Michael Formatted

Michael Deleted: the ...wavelength of 290 nm. The total amount of Fmoc equals to

Michael Because there is an "S" on "acids", you need "number" instead of "amount". You could say "amount of amino acid".

Michael Deleted: amount

Michael Formatted: English (UK)

Michael Deleted: Afterwards... the resin was dried and washed using 3 \times 3 mL DMF and 3 \times 3 mL dichloromethane. The success of the Fmoc deprotection was analysed...d by ...ia a chloranil test. If the test gives the result of green or red resin after 5 minutes of reaction, it this...indicated that the deprotection was successful, but if the test gave...e the result of yellow resin, this it is

Michael This is a nice synonym which means the same thing as "afterwards".

Michael Deleted: Afterwards

Michael Formatted

Michael Deleted: z...d by ...ia a chloranil test. If the test gives ...ave the result of yellow resin after 5 minutes of reaction, it this...indicated that the coupling was successful, but if the test gave...s...the result of green or blue resin, this it...indicated that coupling was unsuccessful and the coupling needed to be repeated. The repetitive deprotection and coupling process was carried out until the desired tetrapeptide attached on...o the resin. Tetrapeptide on the resin was cleaved using 5 mL of 5% of TFA in dichloromethane for 30 minutes for PLPI, PLVI, and PLFI or 5 mL of AcOH:TfE:DCM (2:2:6) for PKAI and PKFI

Michael Because you use "were" I have used the plural here.

Michael Formatted

was collected and concentrated with a rotary evaporator to give purified peptides of PLPI, PLFI, PKAI, and PKFI.

2. Cyclisation of Linear Tetrapeptides and Purification of Cyclic Tetrapeptides

HATU (2 equiv.) was dissolved in 300 µL of DMF, while linear tetrapeptide was dissolved in 10 mL of dichloromethane. Following this, the solution of HATU was added into the solution of tetrapeptide. DIPEA (3 equiv.) was added slowly into the mixture. The mixture of tetrapeptide, HATU, and DIPEA was diluted with 50 mL of dichloromethane. The mixture was stirred at 0 °C constantly for 1 hour. After the mixture stirring under 0 °C, said mixture was continuously stirred at room temperature for 30 minutes. The reaction was monitored via thin layer chromatography using silica GF₂₅₄ nm. The reaction mixture was concentrated using a rotary evaporator after the reaction had finished. Crude of cyclotetrapeptide was dissolved in 5 mL of chloroform and extracted with 1M HCl (3 x 25 mL), saturated NaHCO₃ (3 x 25 mL), and saturated NaCl (3 x 25 mL), respectively. Further, the organic layer was concentrated using a rotary evaporator. The analytical RP-HPLC profile showed that c-PLVI did not need to be further purified. Other crude was then purified using a chromatography column with silica gel as the stationary phase and isocratic n-hexane:ethyl acetate (1:1) as the mobile phase to give purified peptide. The purified c-PLPI, c-PLFI, c-PKAI, and c-PKFI were subjected to analytical RP-HPLC with C-18 column (4 mm x 125 mm, 5 µL) using eluent of water:acetonitrile (5-95%) for 30 minutes, and flow rate of 1 mL/minute for the purity analysis.

3. Side-group Deprotection of Protected-linear and Cyclic Tetrapeptides

A side-group-containing linear and cyclic peptides was deprotected with 95% TFA in water (1 µL of TFA/1 mg of peptide). The reaction mixture was stirred at a slow rate using a Syrris batch reactor for 30 minutes. The reaction mixture was concentrated using a rotary evaporator and dissolved in 25 mL of chloroform before being extracted using 25 mL of saturated NaHCO₃ and 25 mL of brine solution. The organic layer was dried with anhydrous MgSO₄ and the filtrate was collected. The filtrate was concentrated using a rotary evaporator. The crude solution was purified using a chromatography column with silica gel as the stationary phase, and a gradient of n-hexane:ethyl acetate (50-100%) as the mobile phase. Each fraction was collected and concentrated

using a rotary evaporator to give a pure substance containing unprotected linear and cyclic peptides.

Michael	Deleted: z	
Michael	Deleted: . L	
Michael	Deleted: Afterwards	
Michael	Deleted: under	
Michael	Formatted	↓
Michael	Deleted: one	
Michael	Formatted	↓
Michael	Deleted: the	
Michael	Deleted: under	
Michael	Deleted: by	
Michael	Deleted: was	
Michael	Deleted: A	
Michael	Formatted	↓
Michael	Formatted	↓
Michael	Please watch out for little typos like this.	
Michael	Formatted	↓
Michael	Formatted	↓
Michael	Deleted: S	
Microsoft Office...	Deleted:	
Michael	Formatted	↓
Michael	Deleted: of	
Michael	Deleted: were	
Michael	Deleted: with	
Michael	Deleted: and	
Michael	It seems that you mean this....	
Michael	Formatted	↓
Michael	Formatted	↓

Michael Deleted: of